# From symptom onset to treatment initiation: protocol for a narrative study exploring the journey of older adults with tuberculosis in the English Midlands, UK

Farah Kidy ,[1] Noel McCarthy ,[2] Kate Seers [3]

[1]Warwick Medical School, University of Warwick, Coventry, UK
[2]Population Health Medicine, Public Health & Primary Care, Trinity College Dublin, Dublin, Ireland
[3]Warwick Research in Nursing, Warwick Medical School, University of Warwick, Coventry, UK

**Correspondence to**
Dr Farah Kidy;
f.kidy@warwick.ac.uk

## ABSTRACT

**Introduction** Time from symptom onset to treatment initiation in tuberculosis (TB) remains stubbornly prolonged despite reductions in disease incidence. Delays may contribute to increased morbidity, mortality, onward spread of disease and poor patient experiences. Most delays occur prior to hospital referral. The average primary care healthcare provider in England is unlikely to see TB on a regular basis. Little is known about primary care diagnostic and referral challenges.
Adults aged 65 years or older are more likely to experience delays. However, little is known about their journey from symptom onset to treatment initiation.

**Methods and analysis** We will carry out a narrative study including adults aged 65 years or older, living in the English Midlands and receiving treatment for active TB. Twelve English and 12 Urdu or Punjabi speakers will be recruited from TB clinics and interviewed. Their primary care records will be accessed, and the primary care story and secondary care letters will be extracted. Each of the data sources will be analysed using dialogical narrative analysis. Data will be triangulated within participants and across the data set.

**Ethics and dissemination** This study received approval from the Health Research Authority and the Research Ethics Committee in April 2022. Risk management and equity considerations have been made a priority. Findings will be disseminated through publication in open access peer-reviewed journals, presentations to policy makers, primary healthcare and secondary healthcare professionals, and through public facing materials developed in conjunction with patients, members of the pubic, TB services and charities.

## STRENGTHS AND LIMITATIONS OF THIS STUDY

⇒ Using narrative methods allows for an understanding of the sequence and experience of events and opens the door to exploring causal mechanisms.
⇒ This study will recruit participants whose experiences are seldom considered and who are most likely to suffer from negative consequences of tuberculosis.
⇒ Including only Urdu, Punjabi and English speakers is a potential limitation.
⇒ Case notes may be challenging to analyse and interpret.

median delay (79 days, IQR 40–153) has improved in the last 5 years.[1]

The TB Action Plan for England sets out the priorities and actions for TB services, the UK Health Security Agency, the NHS in England and other stakeholders to ensure that the country meets its commitment to eliminate TB as a public health problem by 2035.[2] Recovery from COVID-19 and prevention, detection and control of TB are prioritised. Addressing treatment delays underpins activities within all these areas.

Treatment delays are a particular focus for action because they may lead to harmful consequences including increased morbidity,[3–5] mortality,[3] transmission[3 6 7] and poor patient experiences.[8 9]

Adults aged 65 years or older contribute 16% of cases with pulmonary TB but make greater contribution to those who experience delays of 4 months or longer.[10–16] Older adults are also more likely to develop extrapulmonary disease which is associated with even longer delays (median 9 months in orthopaedic disease).[8 17 18] A greater proportion of this age group require additional support during treatment (37% vs 32.9% of 45–64 year olds, or 28.2% of 15–44 year

## INTRODUCTION

England is a low-incidence tuberculosis (TB) country with an incidence of 7.8 per 100 000 in 2021.[1] Despite decreases in the incidence of TB since 2011, time from symptom onset to treatment initiation (treatment delay) remains stubbornly prolonged. Neither the proportion of people waiting (31.6% waited for 4 months or longer in 2021) nor the

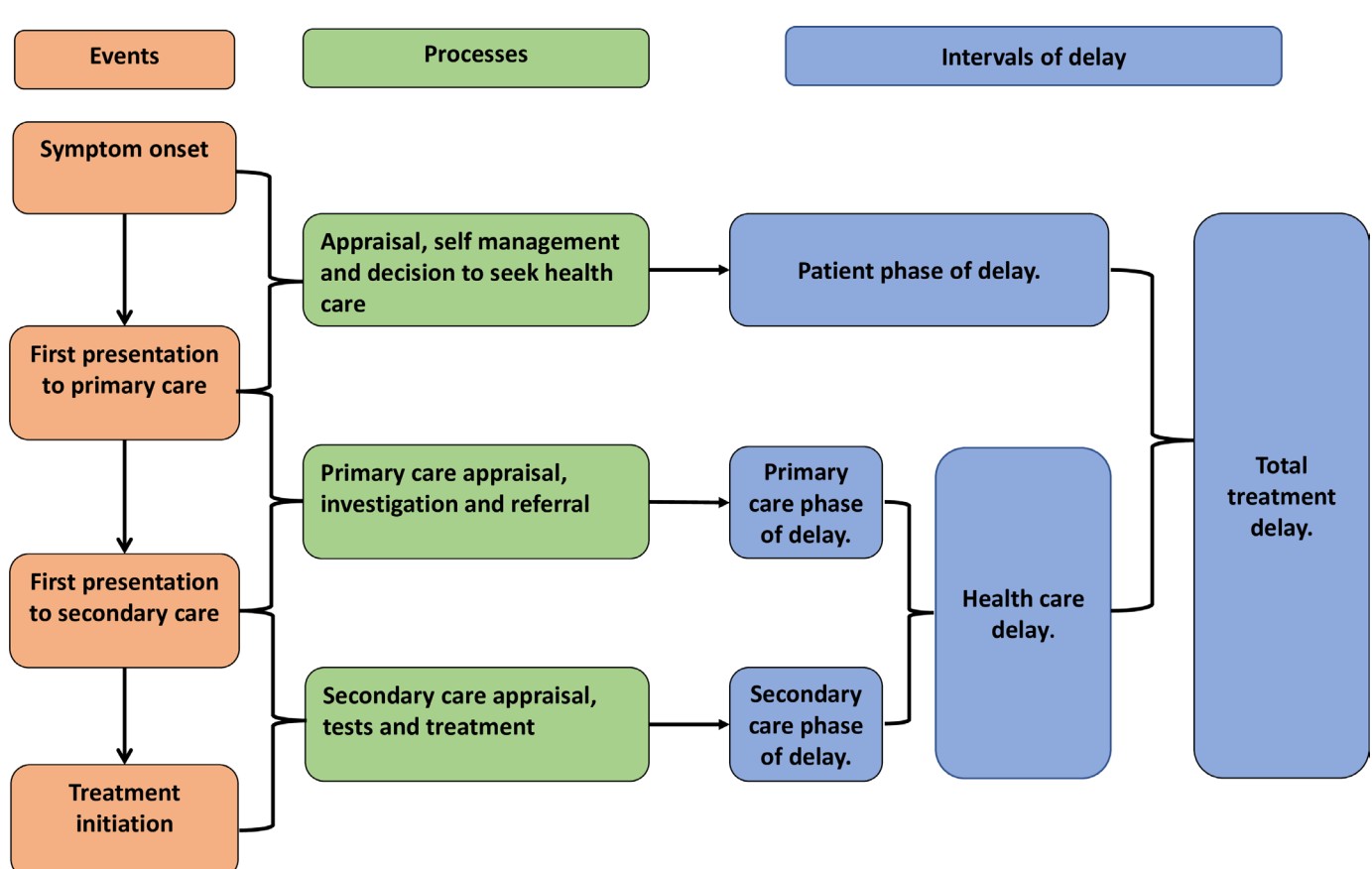

**Figure 1** Showing events, processes and intervals of delay in the patient journey from symptom onset to treatment initiation.

olds).[1] This support ranges from additional contact with TB nurses to directly observed therapy. They are also less likely to complete treatment than their younger counterparts (65.6% vs 84.8%, or 88.3%, respectively).[1]

Research to reduce treatment delays is most advanced in cancer studies. Here a decade of experience highlights the importance of exploring the whole patient journey. From the process of self-appraisal, which identifies that there has been a shift from the norm, to the decision to seek healthcare and subsequent actions of healthcare professionals.[19] A modified version of Walter *et al*'s model provides a framework to reflect on as the study progresses, figure 1.

### The patient phase of delay

There is a body work (using methods such as focus groups,[20] interviews,[9 21 22] and surveys or questionnaires[23–26]) exploring healthcare seeking behaviours in adults with TB in low-incidence countries. Perceived stigma and fear of being isolated from loved ones were barriers,[9 20 23] particularly among migrant populations.[9 21 23 27] In some cases, mild symptoms were misinterpreted as being due to something which would resolve itself or could be treated at home.[9 22 24–26] In some settings, practical considerations such as distance to healthcare facility, having to pay for transport or anticipated wait times were barriers.[23 24 27] Being a new migrant or facing

language barriers delayed presentation to healthcare services.[26 28] Unfortunately, few participants aged 65 or older were included in these studies.

There is a long-standing argument that older adults experience ageism at individual, policy and system levels when engaging with healthcare.[29] The evidence about self-appraisal and healthcare seeking decisions in older adults is mixed. In a review of cancer studies, age was found to be associated with both increased and decreased patient phase delays.[30] The appraisal process was influenced by symptom awareness and interpretation.[30 31] The decision to seek healthcare was influenced by the support of others, independence, awareness of systems, fear of potential diagnoses and self-management including complementary medicine.[30 31] These influences could both prompt and delay seeking healthcare. These reviews did not include participants with TB.

### The healthcare phase of delay

Studies from Europe and the UK have shown greater relative contribution of either patient[32 33] or healthcare[26 34–37] stages of treatment delay. In our context, TB is diagnosed in secondary care settings (ie, outpatient specialist clinics or during an inpatient admission). Most studies did not differentiate between the stages of healthcare delay, but Roberts *et al* report

that most healthcare delay occurred prior to referral to secondary care.[28]

## The knowledge gaps

To influence patient journeys, we need to understand both what happened and why it happened. This knowledge is missing for older adults with TB from patient, primary and secondary care perspectives. The objective of this study is to address these knowledge gaps by exploring:

1. The sequence of events leading from a point of no new illness to the point of treatment initiation.
2. The experience of living through these events.
3. The impact of others (eg, family and healthcare providers) on the sequence and experience of these events
4. Recognition in and referral from primary care
5. Recognition and diagnosis in secondary care.

## METHODS AND ANALYSIS
### Ontology and epistemology

Bhaskar, the founder of Critical Realism, offers a three-level description of reality. The *empirical* level is that which is observable, the *actual* level is the reality which exists irrespective of our observations and about which we may or may not be aware and lastly, the *real* level which encompasses unseen causal influences.[38] We assume that there is an *actual* patient journey from symptom onset to treatment initiation, but that our description and understanding of this will vary depending on the method of inquiry used and to whom this is delivered. Explicit acknowledgement of the *real* level allows us to delve into questions about causal mechanisms and how or when they are triggered within complex systems.[39]

### Methodology

This study will use narrative inquiry applied to interviews and written records. Data are collected in the form of narratives (or stories) and the whole narrative is the unit of analysis. A narrative has a beginning followed by a sequence of unfolding events, culminating with an end.[40] There is a plot and there are characters other than the narrator.[41] Narratives tell us how humans experience and make sense of the world.[42] Listening to the whole story allows us insight not only into what has happened but also why it happened and what that felt like. There is an explicit focus on understanding how the illness fits into patients' lives.[43] Narrative approaches have been used successfully to explore healthcare experiences of older adults[44] including those with a different first language to the investigators.[45]

Narrative methods will work well for our population since they allow participants to tell the story that they think is important when they are neither seeking healthcare nor are there any negative consequences to their stories.

Narrative inquiry fits well within Critical Realism as there is an assumption of an *actual* patient journey with people, structures and events. We are *empirically* describing this from both participants' memories and their health records. We will use the content and structure of the narratives to come to conclusions about the *real* mechanisms underlying healthcare seeking behaviour and arrival at diagnoses.

### Sample size

Some researchers believe that a priori determination of sample sizes is not possible[46] rather that data saturation or data adequacy be used.[47] Others suggest that for non-probabilistic sampling in homogenous populations, most themes are apparent after six interviews and that saturation is reached after 12 interviews.[48] However, Hennink *et al* distinguish between code saturation (heard it all) and meaning saturation (understood it all).[49] With the aim of ensuring meaning saturation of the main narrative themes, we will recruit up to 24 patients, with up to 12 who speak English and up to 12 who speak Urdu or Punjabi.

### Recruitment and eligibility

Participants will be identified from TB clinics in the English Midlands by members of their normal care team.

Patients will be considered eligible if they satisfy all the following criteria:

▶ Aged 65 years or older at the time of diagnosis.
▶ Any gender.
▶ Resident in the catchment area.
▶ English, Urdu or Punjabi speakers.
▶ Receiving treatment for active TB.
▶ Managed in the community.
▶ Able to give informed consent.
▶ Hear well enough to have a remote conversation.

The case definition for active TB is:

1. Culture of a sample has resulted in the growth of any of the *Mycobacterium tuberculosis* complex of organisms, OR.
2. A clinician's judgement that the patient's clinical condition is compatible with TB, AND a clinician's decision to treat the patient with a full course of anti-TB therapy.[50]

Strain of TB, resistance or relapses will not influence eligibility.

Urdu and Punjabi were selected from among other languages as they are the most frequently spoken non-English languages in our catchment area.

A stratified purposive sampling approach will be taken. Participants will first be stratified by language (English or Urdu/Punjabi speaking) and then gender (see figure 2). Within each stratum, there will be purposive sampling to select those with differing sites of disease (pulmonary or extrapulmonary), a range of delay times and a spectrum of ages over 65 years. To avoid bias, outstanding strata for recruitment will be regularly reviewed with the clinical teams.

Participants will be given the option of taking part in the interview with or without the case note review.

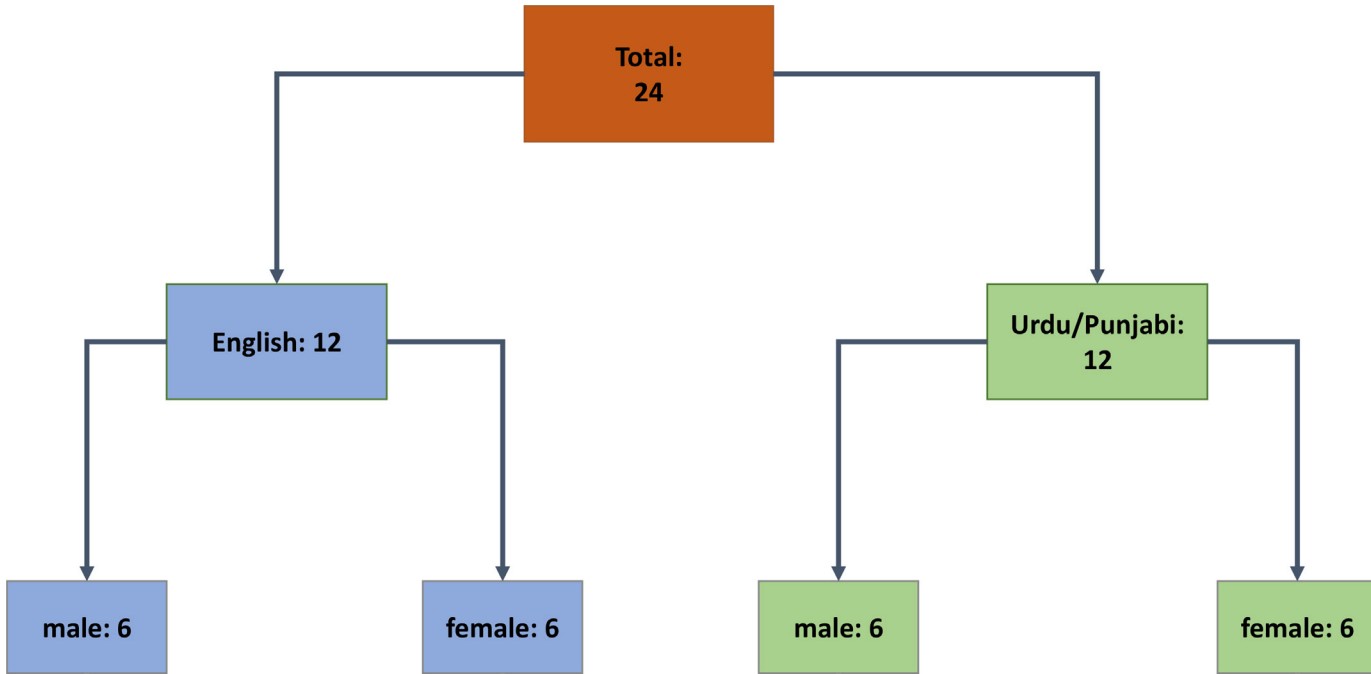

**Figure 2** Showing planned stratification of participants.

Participants will be offered a £20 voucher for taking part in the study.

Recruitment will take place from April to December 2023.

### Data collection

Data will be collected from the participant and from their primary care record generating a patient story, a primary care story as recorded in the notes and a secondary care story as recorded in letters.

#### The interview

Participant experience will be captured using a narrative interview. The interview is expected to last up to one and a half hours, with additional travel time for those being interviewed in person. A modification of the schema for an unstructured, in-depth interview suggested by Jovechelovitch and Bauer will be applied.[51]

1. Initiation of the interview. An open question will be asked which defines the story of interest, its start point and its end point.
2. Main narration. During this phase, there will be minimal interruptions, with simple verbal and non-verbal cues to encourage the participant.
3. Questioning phase. Key events of interest will be identified prior to the interview and the participant's own words will be used to probe further about these events if they have not already been discussed. However, this will not include seeking opinions or judgements.
4. Concluding talk. During this phase, member checking will be carried out by directly asking participants if they are happy with the information shared.

Interviews in English will be carried out by FK. Interviews conducted in Urdu and Punjabi will be carried out by trained, research aware members of university staff and trained postgraduate students. If these individuals are not available, named individuals from formal translation services will be trained and engaged for interviews. All interviews will be supervised by FK. FK has experience in conducting focus groups and has experience of collecting narratives in clinical settings. She has undertaken training in narrative methods.

A topic guide has been developed to ensure that demographic details and personal characteristics (such as employment status, living situation and country of birth) as well as key turning points in the story are captured (online supplementary material 1). This will be tested with one English speaking and one Urdu or Punjabi speaking participant and adapted as needed.

Interviews will be transcribed verbatim in the language of the participant and translated into English where needed. Data will be managed by using NVivo.

#### Primary care records

The eligibility time for extraction from records will start from 12 months prior to the date of symptom onset as reported by participants and will end at the date of treatment initiation as reported by the TB clinic. This long look-back period is chosen since there is limited evidence about errors in recall of onset date.[52]

A list of alert symptoms for TB has been developed for another part of this project. This includes general symptoms such as fever, night sweats and weight loss; site-specific symptoms such as productive cough or enlarged lymph nodes for disease in the lungs or lymph nodes, respectively; and tests commonly carried out in primary care which may be abnormal in TB. Any consultation, within the eligible time, in which these signs, symptoms or TB disease manifestations uncovered during interviews

are mentioned will be extracted. Where a consultation deals with more than one problem, and the TB related entry can be clearly differentiated, then only this entry will be extracted. A frequency count of tests will be noted, but only abnormal tests results will be extracted. Letters to and from secondary care within the eligible time frame which are related to the TB journey will also be extracted.

Relevant consultations will be identified by checking the case notes from the beginning of the eligibility period. This approach was selected as using search functions may miss data where there are misspellings or unusual abbreviations. Date of first presentation, date of referral, frequency of consultations, number of team members consulted, number of secondary care clinics attended and presence of comorbidities will be recorded and used to describe participants.

Data extraction within GP surgeries will be carried out by FK. FK has experience in the use of several different primary care electronic healthcare record systems.

Data collection (interviews and case note reviews) will take place from April to December 2023.

### Cross cultural research

There is cocreation of the narrative between the researcher and the narrator; the story told is influenced by the audience.[53] In this study, for 12 of the participants, interpreters will be gathering data with minimal interruptions to translate the content. To aid this process, interpreters have been recruited from among bilingual, research aware members of the university community. The decision to restrict the range of languages to only Urdu and Punjabi was taken to ensure maximal chances for recruitment and training of appropriately qualified interpreters. We accept that this decision may result in missing information, particularly in ethnic minority groups.

Stories gathered in Urdu or Punjabi will be analysed in English. In order to retain the depth of detail and nuance of the data, we will apply an approach suggested by Lopez et al.[54] Where the interview was conducted by an interpreter, the audio recording will be translated by an independent person with the aim of conceptual equivalence. The second translator will be asked to highlight where idioms have been used and what these mean.[55] A university-based interpreter will be asked to review the written translations and compare these to the audio recordings. Any discrepancies will be resolved by discussion with the study team.

### Data analysis

Narrative analysis refers to a collection of different methods with the shared characteristic of analysing narratives as a whole unit. This group of methods was chosen since they allow investigation of events, the experience of living through those events, the sociocultural contexts of the participants and the way in which the participant has chosen to tell their story. These features are important in achieving the objectives of this study.

The three data sources (interview, primary care record and secondary care letters) for each patient are distinct stories with different narrators and viewpoints. The main analysis will therefore treat each data source separately. Additionally, based on the Critical Realism mindset which sees each data source as a different tool to capture the *actual* patient journey, we will explore the data through triangulation after matching data sources for individual patients.

We will carry out dialogical narrative analysis using a method recommended by Smith.[41] Although this is presented as a numbered list, the process is iterative, and steps two to five will be repeated several times:

1. 'Indwelling'—this is the process of familiarisation by reading the transcript several times.
2. Identifying or smoothing the story—at this stage, the 'core story' or stories are identified. This enables shortening of the narrative to aid further analysis.
   For the interviews, comments from the interviewer will be removed. The text will be rearranged to provide a chronological arc to the story. Extraneous material (if any) will be marked in a different colour.[56]
   For the textual data sources, portions of the consultations or letters which refer to health problems which are clearly different to TB will be removed. Again, data will be arranged chronologically if needed.
3. Identifying narrative themes—rather than line by line coding, the aim here is to identify themes which run through the whole story. At this stage, we are most interested in exploring what is said. Both manifest and latent themes may be present.
4. Identifying the structure of stories—the aim here is to understand 'how the story is put together'.[41] At this stage, we are interested in identifying how things have changed over time. For example, there may be stories of progress or stability over time.
5. Opening up the analysis—at this stage, Smith recommends asking a series of questions of the stories.
   a. Resource questions. Why is the person telling this story in this way? What templates of narratives are they drawing on to tell this story?
   b. Circulation questions. Who is the intended audience?
   c. Connection questions. Does this story create a 'them' and 'us'? Is there someone the storyteller does not want to hear this story?
   d. Identity questions. Does the story give the person a sense of identity? Or does it identify who they might want to be in the future?
   e. Function questions. Does the story help or hinder the storyteller? What effect can or could the story have on other people?

Twenty per cent of the data will also be analysed by KS to ensure that no core stories or narrative themes are missed.

The results will be drawn together by building a typology of stories. Stories from within each data source and across the whole data set will be clustered together according

to shared characteristics in the domains of themes, structure, resources and function.

## Triangulation

The approach to triangulation will follow a method suggested by Farmer *et al*.[57] Data will be triangulated within participants and across the whole data set by looking for convergence, completeness, silence and dissonance across the domains of narrative themes, story structure, resources and function. This will allow us to understand how the perspectives of the story tellers impact on the story told.

Analysis will be completed by March 2024.

## Bringing rigour

Trustworthiness and plausibility[58] will be evidenced through member checking at the end of the interview by asking participants if they are happy with all the material they have shared. The context of interviews and an explanation of data gathering techniques will also be provided in the report.

In order to increase criticality,[58] FK will keep a contemporaneous reflexive diary to help understand her role as a researcher rather than GP, and to explore her own biases and preconceptions.

## Patient and public involvement

People with living TB, older adults with chronic conditions and their carers were consulted about the project aims, methods and implementation. This directly influenced the decision to interview participants alone, impacted on the locations for interviews and the methods of dissemination. Patient partners commented on the information leaflets and consent forms. Bilingual members of the public helped to recruit translators.

## ETHICS AND DISSEMINATION
## Ethics

Some of the key ethical issues raised by the study are discussed below.

## Equity

TB often affects the most vulnerable in society such as those living in deprivation or experiencing the effects of substance misuse or time in prison.[16] It is important that we include these voices in our research. TB teams have been sensitised to identification of participants based on the eligibility criteria alone. If there are concerns about violence towards staff or difficulty engaging, participants will be offered remote interviews only.

TB affects migrant populations. Our approach to recruitment means that English speaking migrants from a range of backgrounds can be included. However, we have restricted the number of languages included and this means that some voices will not be heard.

We have made the decision to transcribe all interviews in the language of the participants to allow reporting in the source language and to ensure that all participants' original words are available for secondary analysis.

## Recruitment

The duration of TB treatment means that clinical teams and patients may develop strong relationships. To reduce the risk of participants feeling obliged to participate, they will be made aware that non-participation will not negatively affect their care and that they are free to withdraw at any time. Although the clinical teams will identify potential participants, consent will be obtained by the research team.

## Harms to participants

Some participants may find it distressing to talk about their experiences. Should this occur, the interview will be paused or terminated. FK is an experienced clinician and has training and experience in de-escalating situations. The nature of narrative interviews means that participants are in control of the level of detail they reveal. Participants will be allowed time to debrief and will be signposted to local support services if required.

Interviews will take place remotely or in clinical spaces and during normal working hours to limit potential environmental harms to both participants and researchers, and to allow prompt responses to expressed clinical needs. Input from our PPI partner suggested that evenings and weekends were valuable free time for those in employment and so should not be used for interviews. This approach may result in recruitment bias against those who are in full-time employment. We will report the numbers of those who were unable to take part in the study due to this decision.

## Confidentiality

Data will be stored in anonymised, encrypted, electronic formats. IT Security Teams have been consulted to ensure that data storage and transfer procedures are appropriate. Aspects of the narrative which might lead to identification of participants will not be reported.

The patient information leaflet clarifies that there may be situations in which disclosures are needed. In line with General Medical Council, UK, guidelines,[59] participants will first be asked for permission to disclose information. If permission is not forthcoming, then identifiable information will only be shared if this is in the interests of the patient or the public. All disclosures will be noted as part of the participants research record.

## Stigma

Erving Goffman described three domains types of stigma, that associated with physical or mental illness (eg, TB), that associated with identifying as part of a marginalised group (eg, ethnic minorities or being an older adult) or that associated with moral character (eg, substance misuse) (quoted in[60]). Our study participants are vulnerable to stigmatisation due to a confluence of one or more of these domains. Following guidelines suggested by Gabbidon and Chenneville,[60] we are reducing inadvertent

stigmatisation by engaging in reflexivity around positionality and consulting with patient partners. Narrative methods give control of the content shared to study participants before allowing probing by the interviewer. We have clarified pathways for additional support should the interviews cause distress. We intend to describe the context and will take care to avoid identification through quotes. Participants will be anonymously acknowledged.

### Infection control concerns

Face to face interviews will only take place after the TB team confirm that this is safe. The use of personal protective equipment and distancing measures will be determined by national guidelines, University and NHS Trust policies.

### Case note reviews

Accessing case notes may make participants feel exposed or vulnerable. To this end, participants will be able to consent to the interview study alone.

Data extraction will be kept proportionate by using a preidentified list of symptoms and signs or the participant's own words to identify appropriate material for the study.

Case note review may also make participants' GPs feel exposed. A graded approach will be taken to handling potential risks to participant safety. Where there is an ongoing or active risk, for example, monitoring blood tests not being carried out, this will be flagged up using an auditable electronic messaging service.

Where there is the suggestion of historic risk to patient safety, for example, avoidable treatment delay, the surgery will be made aware of this as a potential significant event.

Preferred routes and recipients of each type of communication will be determined on a surgery-by-surgery basis.

### Dissemination

This study forms part of a larger body work which will be submitted and published as a PhD thesis. Following analysis, a plain language summary will be shared with participants. Findings will be shared in peer-reviewed journals and through public facing materials developed in conjunction with TB services, primary care colleagues, patient partners and TB charities.

**Contributors** FK conceived the study with support from KS and NM. FK wrote the first draft of the paper and all authors commented on the revisions.

**Funding** FK is funded by the NIHR Doctoral Research Fellowship, grant number 300688. The views expressed are those of the author(s) and not necessarily those of the NIHR or the Department of Health and Social Care. The project is being delivered with support from the Clinical Research Networks for Primary and Secondary Care in the West Midlands.

**Competing interests** None declared.

**Patient and public involvement** Patients and/or the public were involved in the design, or conduct, or reporting, or dissemination plans of this research. Refer to the Methods section for further details.

**Patient consent for publication** Not applicable.

**Ethics approval** This study involves human participants and was approved by the study was approved by the London—Harrow Research Ethics Committee.

REC reference—22//PR/0357 Protocol number SOC.06/20-21IRAS ID 287971. Participants gave informed consent to participate in the study before taking part.

**Provenance and peer review** Not commissioned; externally peer reviewed.

**ORCID iDs**
Farah Kidy http://orcid.org/0000-0003-0771-5052
Noel McCarthy http://orcid.org/0000-0003-1113-1017
Kate Seers http://orcid.org/0000-0001-7921-552X

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
