## [Reviewer comments · BMJ Open]

ARTICLE DETAILS

TITLE (PROVISIONAL)	From symptom onset to treatment initiation: protocol for a narrative study exploring the journey of older adults with tuberculosis in the English Midlands, UK
AUTHORS	Kidy, Farah; McCarthy, Noel; Seers, Kate

VERSION 1 – REVIEW

REVIEWER	Jansa, Josep Maria European Centre for Disease Prevention and Control, Epidemic Intelligence and Response
REVIEW RETURNED	03-May-2023

GENERAL COMMENTS	Line 21 in the Southeast of England, most health care delay occurred prior to referral to secondary care.[21]. if Secondary care means hospitalization or another in the healthcare process, this should be specified. Line 47: This study forms part of a larger, mixed-methods research project exploring treatment delays in TB, its causes, and its consequences. Please include reference Comments to authors Dear colleagues, please consider the following topics for a better understanding of the study protocol. 1. Total estimated time for the study, since the recruitment to the final analysis2. Please include a definition or proper reference for active TB.3. Are the selected participants sensitive to all first line TB drugs? Do you include relapses, MDR or XMDR patients?4. Targeted professionals (you stated, “presentations to policy makers”). Please include primary health care and hospital health care professionals.5. Total time needed for participants if any, before and after the interview.6. Characteristics of the participants.a. Countries of origin for English speakersb. Number of man and women according to TB prevalence and gender characteristicsc. Socio economical profiles7. Better explain the overall context of the research
--

REVIEWER	Loutet, Miranda University of Toronto Dalla Lana School of Public Health, Epidemiology
REVIEW RETURNED	13-May-2023

GENERAL COMMENTS

Overall, this protocol paper is well written and thoroughly describes the methods of the study. This is an important study and it's great to see the use of qualitative methods to conduct a deeper dive into the intricacies of this complex public health challenge. There are a few minor clarifications to the rationale and methods that I have detailed below.

Introduction

Page 3, line 59-60: you say "nor the mean delay" but state the median in the brackets, so choose either mean or median, whichever appropriate for the data.

Page 4, line 3-5: integrate the definition of treatment delay into a sentence where using the term. It feels awkward stated as it is now as a sentence on its own.

Page 4, lines 20-23: add in some context of the epidemiology of TB among older people in the UK – incidence, treatment rates, etc. this will strengthen the rationale for why this population was selected. Also provide more detail for what you mean by "disproportionately affected", we need to see a stronger rationale for why this population is the focus of your study.

Page 4, lines 46-58: this is a very good summary of the literature on delays from the patient-perspective, but because one of the novel aspects of this project is the use of narrative methods, it would be useful to summarize what other methods have been used to study this topic among patients with TB.

Methods

General questions/comments:

- No dates for the proposed study were included.
- Please provide details on who will be collecting data (interviews and extraction from patient records) and what their level of training will be.
- Will other sociodemographic information be collected on the participants such as marital status, employment, educational background? If so, please report what information will be collected and from what source.

Page 5, line 47-48: authors say this protocol is part of a larger project – has the protocol or results for that project been published? If so, reference it here. If not, then provide more details about the larger study.

Page 6, lines 14-16: this is important to know that narrative approaches have been used successfully in similar study populations, can you provide more detail as to why you think they work well for your study population?

Ethics and dissemination

General comments:

- A very important public health challenge with the care of patients with TB is stigma. The authors have mentioned ways that they will minimize stigma but do not specifically discuss that these methods are to target stigma. Please include a discussion that shows the authors have specifically thought about how stigma may occur and mitigation strategies.

	Page 10, line 23: please refrain from using stigmatizing language like “chaotic lifestyles.” Instead refer to known risk factors and social determinants of TB. Page 10, lines 31-32: please expand on your rationale for only including the languages spoken by the majority of the population and the potential biases that this will have for your results. Page 10, lines 55-56: please provide rationale for only having interviews during working hours as this might negatively impact participants who themselves work and may create bias in your results.
--	--

VERSION 1 – AUTHOR RESPONSE

Reviewer one:

Comment	Response
Line 21 in the Southeast of England, most health care delay occurred prior to referral to secondary care.[21if Secondary care means hospitalization or another in the healthcare process, this should be specified.	We have clarified that secondary care refers to inpatient or outpatient hospital care by adding the following sentence: “In our context, TB is diagnosed in a secondary care setting (ie outpatient hospital specialist clinic or during an admission).”
Line 47: This study forms part of a larger, mixed-methods research project exploring treatment delays in TB, its causes, and its consequences. Please include reference	This study is part of a series of work packages leading to a PhD qualification. Details of each work package have not been published, but a summary is available on the funder’s website and this has been referenced.
Total estimated time for the study, since the recruitment to the final analysis	Thank you for pointing this out. The following has been added. “Recruitment and interviews will take place from April to December 2023 and analysis will be completed by March 2024.”
Please include a definition or proper reference for active TB.	The definition applied by public health authorities in England has been added to the report.
Are the selected participants sensitive to all first line TB drugs? Do you include relapses, MDR or XMDR patients?	We will not be using drug sensitivities to determine eligibility. So, if patients with relapses or patterns of resistance agree to take part, they will be included. We have clarified this in the eligibility criteria.
Targeted professionals (you stated, “ presentations to policy makers ”). Please include primary health care and hospital health care professionals.	Thank you for this reminder. Health care staff have now been included.

Total time needed for participants if any, before and after the interview.	We have added the following detail: “The interview is expected to last up to one and a half hours, with additional travel time for those being interviewed in person.”
Characteristics of the participants. a. Countries of origin for English speakers b. Number of man and women according to TB prevalence and gender characteristics c. Socio economical profiles	Thank you for this suggestion. We will collect data on the sex of participants, age at diagnosis, time to diagnosis and site of disease on our first consent form. We will add questions to interview to understand country of origin, living situation and employment status.
Better explain the overall context of the research	We have added additional detail about the epidemiology of TB in general and that of older adults in particular to the introduction. We have included another figure (the new Figure 1) to highlight how much more older adults are affected by treatment delays. “England is a low-incidence tuberculosis (TB) country with an incidence of 7.8 per 100,000 in 2021.[1] Despite year-on-year decreases in the incidence of tuberculosis (TB)TB in England since 2011, time from symptom onset to treatment initiation (treatment delay) remains stubbornly prolonged.” “Adults aged 65 years or older contribute approximately 16% of cases but make a far greater contribution to those who experience delays of 4 months or longer, Fig 1.[10-16] These data only show cases with pulmonary TB. are disproportionately affected by treatment delays when they have disease in the lungs.[16] Older adults are also more likely to develop extra-pulmonary disease which is associated with even longer delays (median 9 months in orthopaedic disease).[8, 17, 18] A greater proportion of this age group require additional support during treatment (37% vs 32.9% of 45 to 64 year olds, or 28.2% of 15 to 44 year olds) and they are less likely to complete treatment than their younger counterparts (65.6% vs 84.8%, or 88.3%, respectively).[1]”

Reviewer two:

Comment	Response
Overall, this protocol paper is well written and thoroughly describes the methods of the study.	Thank you very much for your kind remarks.

This is an important study and it's great to see the use of qualitative methods to conduct a deeper dive into the intricacies of this complex public health challenge. There are a few minor clarifications to the rationale and methods that I have detailed below.	
Page 3, line 59-60: you say “nor the mean delay” but state the median in the brackets, so choose either mean or median, whichever appropriate for the data.	We have changed the sentence to read “median delay”
Page 4, line 3-5: integrate the definition of treatment delay into a sentence where using the term. It feels awkward stated as it is now as a sentence on its own.	We have integrated the definition into an earlier sentence.
Page 4, lines 20-23: add in some context of the epidemiology of TB among older people in the UK – incidence, treatment rates, etc. this will strengthen the rationale for why this population was selected. Also provide more detail for what you mean by “disproportionately affected”, we need to see a stronger rationale for why this population is the focus of your study.	We have included further contextual and epidemiological detail to the introduction and have included a figure (the new figure 1) to highlight what we mean by older adults being disproportionately affected. “Adults aged 65 years or older contribute approximately 16% of cases but make a far greater contribution to those who experience delays of 4 months or longer, Fig 1.[10-16] These data only show cases with pulmonary TB. are disproportionately affected by treatment delays when they have disease in the lungs.[16] They Older adults are also more likely to develop extra-pulmonary disease which is associated with even longer delays (median 9 months in orthopaedic disease).[8, 17, 18] A greater proportion of this age group require additional support during treatment (37% vs 32.9% of 45 to 64 year olds, or 28.2% of 15 to 44 year olds) and they are less likely to complete treatment than their younger counterparts (65.6% vs 84.8%, or 88.3%, respectively).[1].”
Page 4, lines 46-58: this is a very good summary of the literature on delays from the patient-perspective, but because one of the novel aspects of this project is the use of narrative methods, it would be useful to summarize what other methods have been used to study this topic among patients with TB.	A brief summary has been included. “There is a body work (using methods such as focus groups[20], interviews[9, 21, 22], surveys or questionnaires[23-26]) exploring health care seeking behaviours in adults with TB in low incidence countries”
No dates for the proposed study were included.	Thank you for pointing this out. The dates have been added: “Recruitment and interviews will

	take place from April to December 2023 and analysis will be completed by March 2024.”
Please provide details on who will be collecting data (interviews and extraction from patient records) and what their level of training will be.	This information has now been added for both data sources.
Will other sociodemographic information be collected on the participants such as marital status, employment, educational background? If so, please report what information will be collected and from what source.	Thank you for this suggestion. We will add questions to interview to understand country of origin, living situation and employment status.
Page 5, line 47-48: authors say this protocol is part of a larger project – has the protocol or results for that project been published? If so, reference it here. If not, then provide more details about the larger study.	This study is part of a series of work packages leading to a PhD qualification. Details of each work package have not been published, but a summary is available on the funder’s website and this has been referenced.
Page 6, lines 14-16: this is important to know that narrative approaches have been used successfully in similar study populations, can you provide more detail as to why you think they work well for your study population?	We have added: “Narrative methods will work well for our population since they allow participants to tell the story that they think is important when they are neither seeking healthcare nor are there any negative consequences to their stories.”
A very important public health challenge with the care of patients with TB is stigma. The authors have mentioned ways that they will minimize stigma but do not specifically discuss that these methods are to target stigma. Please include a discussion that shows the authors have specifically thought about how stigma may occur and mitigation strategies.	We have added a separate subsection within the Ethics section to detail how stigma may occur and how our approaches can help to mitigate it.
Page 10, line 23: please refrain from using stigmatizing language like “chaotic lifestyles.” Instead refer to known risk factors and social determinants of TB.	Thank you for raising this important point. The phrase has been replaced with: “such as those living in deprivation or experiencing the effects of substance misuse or time in prison”
Page 10, lines 31-32: please expand on your rationale for only including the languages spoken by the majority of the population and the potential biases that this will have for your results.	This decision was taken since the interpreter will also have to act as the data collector. We will not be asking for sentence-by-sentence translation during the interview as we do not want to interrupt the flow of the story. In order to ensure that there is a consistently high level of translation and data gathering, we recruited bilingual members of university staff and provided training in interview skills. The following sentences have been added in the section on cross cultural research: “The decision to restrict the range of languages to only Urdu

	and Punjabi was taken to ensure maximal chances for recruitment and training of appropriately qualified interpreters. We accept that this decision may result in missing information.”
Page 10, lines 55-56: please provide rationale for only having interviews during working hours as this might negatively impact participants who themselves work and may create bias in your results.	The decision to interview during working hours was taken after carrying out a risk assessment and having a discussion with our PPI partner. It was felt that lone working protocols (needed out of hours) would be complex to enact given the geographical breath of the area and that we needed to be able to seek TB Nurse or GP input in case of urgent queries. Equally our PPI partner suggested that free time in the evenings and weekends are also valuable for those who work. You raise a good point about potential recruitment bias. We will report and reflect on how many people are unable to take part in the study due to this decision and whether this results in recruitment bias. We have expanded the explanation to say: “Interviews will take place remotely or in clinical spaces and during normal working hours to limit potential environmental harms to both participants and researchers, and to allow prompt responses to expressed clinical needs. Input from our PPI partner suggested that evenings and weekends were valuable free time for those in employment and so should not be used for interviews. This approach may result in recruitment bias against those who are in full time employment. We will report the numbers of those who were unable to take part in the study due to this decision.”

VERSION 2 – REVIEW

REVIEWER	Jansa, Josep Maria European Centre for Disease Prevention and Control, Epidemic Intelligence and Response
REVIEW RETURNED	11-Aug-2023
GENERAL COMMENTS	The comments from the previous review have been properly addressed.
REVIEWER	Loutet, Miranda University of Toronto Dalla Lana School of Public Health, Epidemiology
REVIEW RETURNED	18-Aug-2023

GENERAL COMMENTS

Thank you for your responses and revisions made to the manuscript in response to my review. Most have adequately addressed my comments, I just have a few follow-up comments to the changes that were made (reported below in the order that they were responded to in the response document).

Introduction:

Page 3, lines 11-20: the added paragraph definitely helps by providing context to the chosen study population but it is unclear how Figure 1 adds to the text. Is the figure meant to show the % of TB cases in each age group that experienced treatment delays? This should be very clear and explicitly said in the title of the figure. Also where is this figure from? Please ensure it is correctly referenced. I defer to the Editor to determine if including a figure from another paper is appropriate here, but I would not think it is conventional or necessary given the other data provided in the text.

Also suggest improving the flow of this paragraph by removing this sentence “These data only show cases with pulmonary TB.” and instead specifying that you are talking about “16% of pulmonary TB cases” in the sentence before.

Also what is meant by “additional support”? Please be as specific as possible with what was measured.

Methods:

Page 9, lines 27-28: thank you for adding the dates of when they study will take place but the chosen section where it was added does not really match to “Bringing rigour.” There are a few options: 1. have a short “Study design” section at the beginning of the Methods that provides a summary of the study design, including the dates of when the study will take place; or 2. move the dates for each phase to the relevant section i.e., recruitment, data collection and analysis.

Page 6, lines 52-53: thank you for adding details of personnel who will be collecting data, but you are still missing who will be conducting in Urdu and Punjabi. Is it trained study staff or university students? If so, it’s ok to just mention that and not specify their names.

Although the authors said they add questions to interview to understand country of origin, living situation and employment status, they have not included this in the protocol paper. It is customary to specify – at least at a high-level (i.e., general themes) – what variables will be collected.

Page 4, lines 44-45: it is not very conventional to call a PhD project a “larger, mixed methods project” in a published protocol paper, referring to it as such makes one think that this particular study is embedded within something like an observational cohort study. I would suggest removing this type of terminology from the peer-reviewed protocol paper (it fits better within a REB submission or PhD thesis).

Page 7, line 47: thank you for adding this rationale for choosing English, Urdu and Punjabi as the primary languages; however, it would be important to add that you acknowledge this may result in missing information from minorities (not just missing information in general).

VERSION 2 – AUTHOR RESPONSE

Reviewer: 1

Dr. Josep Maria Jansa, European Centre for Disease Prevention and Control

Comments to the Author:

Comment	Response
The comments from the previous review have been properly addressed	Many thanks

Reviewer: 2

Dr. Miranda Loutet, Public Health England,

Comments to the Author:

Thank you for your responses and revisions made to the manuscript in response to my review. Most have adequately addressed my comments, I just have a few follow-up comments to the changes that were made (reported below in the order that they were responded to in the response document).

Comment	Response
Page 3, lines 11-20: the added paragraph definitely helps by providing context to the chosen study population but it is unclear how Figure 1 adds to the text. Is the figure meant to show the % of TB cases in each age group that experienced treatment delays? This should be very clear and explicitly said in the title of the figure. Also where is this figure from? Please ensure it is correctly referenced. I defer to the Editor to determine if including a figure from another paper is appropriate here, but I would not think it is conventional or necessary given the other data provided in the text.	The figure was created by extracting data from annual TB reports and was intended to show the percentage of cases experiencing delays by each group. Earlier versions have appeared in the grant application and internal university reports, but not in any published works. We do plan on presenting this figure in the thesis and other published works, so will remove from this protocol. Your comments about improving the clarity of labelling are appreciated and will be carried forward.
Also suggest improving the flow of this paragraph by removing this sentence “These data only show cases with pulmonary TB.” and instead specifying that you are talking about “16% of pulmonary TB cases” in the sentence before.	This change has been made.
Also what is meant by “additional support”? Please be as specific as possible with what was measured.	These data are referring to all levels of enhanced case management. A brief description has been added to the paper.
Page 9, lines 27-28: thank you for adding the dates of when they study will take place but the chosen section where it was added does not really match to “Bringing rigour.” There are a few options: 1. have a short “Study design”	We agree. We have opted to move the phrases into each section.

section at the beginning of the Methods that provides a summary of the study design, including the dates of when the study will take place; or 2. move the dates for each phase to the relevant section i.e., recruitment, data collection and analysis.	
Page 6, lines 52-53: thank you for adding details of personnel who will be collecting data, but you are still missing who will be conducting in Urdu and Punjabi. Is it trained study staff or university students? If so, it's ok to just mention that and not specify their names.	We have added the following. "Interviews conducted in Urdu and Punjabi will be carried out by trained, research aware members of university staff and trained post-graduate students. If these individuals are not available, named individuals from formal translation services will be trained and engaged for interviews. All interviews will be supervised by FK." There is further detail in the cross-cultural research section to explain these decisions.
Although the authors said they add questions to interview to understand country of origin, living situation and employment status, they have not included this in the protocol paper. It is customary to specify – at least at a high-level (i.e., general themes) – what variables will be collected.	We have added this information to the protocol and have updated the topic guide as well.
Page 4, lines 44-45: it is not very conventional to call a PhD project a "larger, mixed methods project" in a published protocol paper, referring to it as such makes one think that this particular study is embedded within something like an observational cohort study. I would suggest removing this type of terminology from the peer-reviewed protocol paper (it fits better within a REB submission or PhD thesis).	This and the associated sentences have been removed.
Page 7, line 47: thank you for adding this rationale for choosing English, Urdu and Punjabi as the primary languages; however, it would be important to add that you acknowledge this may result in missing information from minorities (not just missing information in general).	The phrase "particularly in ethnic minority groups" has been added.

VERSION 3 – REVIEW

REVIEWER	Loutet, Miranda University of Toronto Dalla Lana School of Public Health, Epidemiology
REVIEW RETURNED	03-Oct-2023

GENERAL COMMENTS	the comments from the previous review have been addressed. no further comments.
---